**Subject Area:**
cellular biology

actin, bending penalty, alignment, geometry, gene expression, soft matter physics

**Authors for correspondence:**
Richard K. Assoian
e-mail: assoian@pennmedicine.upenn.edu
Kathleen J. Stebe
e-mail: kstebe@seas.upenn.edu

# Cellular sensing of micron-scale curvature: a frontier in understanding the microenvironment

Richard K. Assoian[1,3], Nathan D. Bade[2], Caroline V. Cameron[2] and Kathleen J. Stebe[2,3]

[1]Department of Systems Pharmacology and Translational Therapeutics, [2]Department of Chemical and Biomolecular Engineering, and [3]Center for Engineering MechanoBiology, University of Pennsylvania, Philadelphia, PA 19104, USA

RKA, 0000-0001-7635-6204

The vast majority of cell biological studies examine function and molecular mechanisms using cells on flat surfaces: glass, plastic and more recently elastomeric polymers. While these studies have provided a wealth of valuable insight, they fail to consider that most biologically occurring surfaces are curved, with a radius of curvature roughly corresponding to the length scale of cells themselves. Here, we review recent studies showing that cells detect and respond to these curvature cues by adjusting and re-orienting their cell bodies, actin fibres and nuclei as well as by changing their transcriptional programme. Modelling substratum curvature has the potential to provide fundamental new insight into cell behaviour and function *in vivo*.

## 1. Introduction

For many years, much of the work in modern cell and molecular biology has focused on understanding how cells sense and respond to cues in their microenvironment. An early and still ongoing approach examined the effects of soluble extracellular signals like hormones, growth factors and cytokines. More recently, research in this area has expanded to include the chemical and mechanical (stiffness) cues in the extracellular matrix (ECM), the insoluble substratum on which most cell types attach and spread. These soluble and insoluble signals have widespread effects on signalling, gene expression, proliferation, motility, differentiation and survival. Nevertheless, most of these studies have been performed with cells cultured on planar surfaces even though curved surfaces, with radii of curvature ranging from the size of a cell (approx. 20–100 microns) to millimetres, predominate in most biological systems.

Membrane proteins use BAR (Bin, Amphiphysin, Rvs) domains to interact with and create membrane curvature on the nano-scale, a topic that has been well reviewed [1–3]. In contrast, less is known about how cells and protein sensors interact with micron-scale curvatures, features particularly prominent in the vasculature, glands and villi. Cells also interact with artificial features on this length scale in the form of implanted biomedical devices for regenerative medicine; might the curvatures in these devices need to be considered during their fabrication? Though the field is nascent, studies to date suggest that micron-scale substratum curvature represents a new frontier in understanding cellular responses to their microenvironment.

## 2. Basics of curvature

The curvature at any point along a curved contour is given by

$$\kappa = \frac{1}{R_c},$$

(2.1)

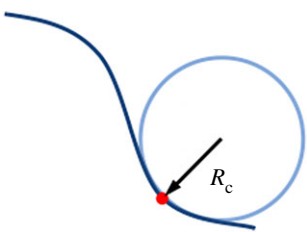

**Figure 1.** Contour curvature. Osculating circle (light blue) at the point of interest (red dot) along a curve (dark blue). $R_c$ = radius of curvature.

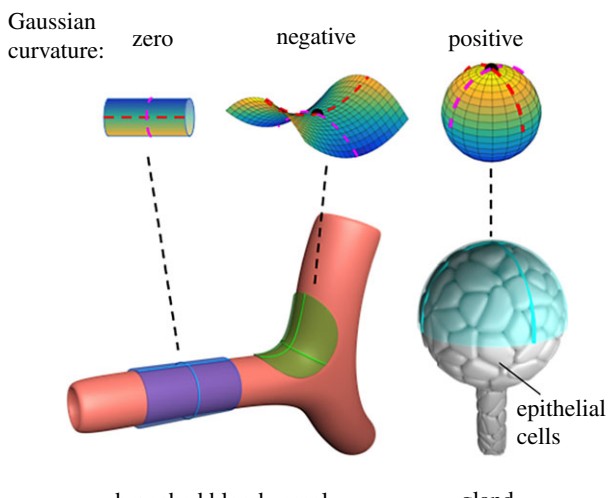

**Figure 2.** Surfaces with various Gaussian curvatures. Top from left to right: representative cylindrical, saddle and bowl/spherical surfaces with zero, negative and positive Gaussian curvature, respectively. Dashed red lines indicate the principal axes of curvature. Bottom: Tissues exhibiting each of these kinds of Gaussian curvature. The cylindrical portion of blood vessels (bottom left) have zero Gaussian curvature (blue), but negative Gaussian curvature exists where one blood vessel branches from another (green). Glands (bottom right) have positive Gaussian curvature (cyan). The two solid lines in each region represent the principal contours of the surface along the principal directions.

where $R_c$ is the radius of an osculating circle at that corresponding point (figure 1). This radius is called the radius of curvature and is the curvature length scale.

The curvature at any point along a surface is characterized by the principal curvatures of the surface at that point (figure 2; see red dashed lines in top images); the principal curvatures $\kappa_1$ and $\kappa_2$ are the maximum and minimum curvatures of surface contours that pass through the point. Surfaces are characterized by various kinds of curvature. For example, the mean curvature is defined as

$$H = \frac{\kappa_1 + \kappa_2}{2} \tag{2.2}$$

and, of particular interest in this review, the Gaussian curvature is

$$K = \kappa_1 \kappa_2. \tag{2.3}$$

The sign of the Gaussian curvature describes the general shape of a curved surface. Surfaces with zero Gaussian curvature must have at least one principal curvature equal to zero, so that the contour tangent to the surface in the principal direction is a straight line (figure 2, left). Examples include planes or

royalsocietypublishing.org/journal/rsob Open Biol. 9: 190155

cylinder-like structures. Saddle-like structures (figure 2, middle) have two non-zero principal curvatures that curve in opposite directions and thus have opposite signs and negative Gaussian curvature. Hill- and valley, or bowl-like, surfaces (figure 2, right) have non-zero principal curvatures that curve in the same direction and thus have positive Gaussian curvature.

## 3. Micron-scale curvature in physiology

The vasculature is a classic example of a complex geometric surface. In mammals, vessel diameters decrease with distance from the heart. The smallest vessels, which can have radii as small as 3 μm [4], consist of endothelial cells and their basement membrane. In fact, the endothelial cells in these small vessels constitute the curved surface itself. Larger vessels incorporate additional layers of cells, either mural cells or vascular smooth muscle cells (VSMCs), depending on the vessel. In large arteries, layers of VSMCs are spaced between layers of elastin. Arterial VSMCs are highly elongated and typically align circumferentially or in a helical direction with a pitch of 20–40° [5]. Cells in the vasculature thus experience zero Gaussian curvature in the straight, cylindrical portions of vessels and negative Gaussian curvature at branch points (figure 2, top middle and bottom left). Epithelial cells, which form many surfaces within the body, including the skin, ducts, the lining of mammary glands and intestinal crypts, and lung alveolae, also constitute and experience curved surfaces on the micron scale. Myoepithelial cells [6,7] interact with glands that have positive Gaussian curvature (figure 2, top and bottom right). Unlike VSMCs, which wrap around blood vessels in the circumferential direction, breast myoepithelial cells align axially along the cylindrical ducts formed by luminal epithelial cells [8]. Similar geometries and curvatures also exist in simpler eukaryotes and throughout the plant kingdom.

## 4. Cellular responses to micron-scale grooves, adhesive islands and curves

Several studies have shown that micron-scale geometric cues influence cell alignment. For example, cells cultured on surfaces with micron-scale grooves align along the long groove axis, and the alignment strength is inversely proportional to groove width (reviewed in [9,10]). Additionally, sharp corners, which have high curvature, alter migration and actin microfilament formation [11]. Cells cultured on microcontact-printed stripes also align strongly in the stripe direction [12,13], showing that sharp corners are not required for contact guidance.

In addition to affecting the alignment of cells, micron-scale geometric cues direct the alignment of actin stress fibres (SFs). SFs are macromolecular bundles of filamentous actin (f-actin) and type II myosins; their development is controlled by a number of actin-binding and -modifying proteins [14–16]. SFs play critical roles in generating cell contractility and are often categorized as ventral, dorsal, transverse arc, and perinuclear actin caps. This categorization is based, at least in part, on the association of SFs with focal adhesion proteins, their position relative to the nucleus and their mechanical properties [14–18].

SFs are responsive to geometric cues. For example, SFs respond to the shape of adhesive islands of various geometries

that are micro-contact printed on planar surfaces. Cells cultured on V-shaped islands form SFs along the periphery of the shape, but also form very prominent SFs that span across the non-adhesive side [19]. When cultured on a complete outlined triangle of the same size as a V-shaped island, cells form more homogeneous SFs than those formed on the V-shaped island [19]. SFs tend to form where an adhesive island has concave edges, and lamellipodia-like structures form where the island has convex corners [20]. Similar results have been reported for advancing Madin–Darby canine kidney (MDCK) epithelial cells cultured in flower-shaped adhesive structures with convex and concave domains [21]. Furthermore, at convex edges the lamellipodia-like structures in these advancing MDCK epithelial cells have the usual retrograde flow of actin fibres, while at convex edges, actin flow is anterograde [21]. Interestingly, by culturing cells on a crossbow-shaped pattern, the morphology of a migrating, cone-shaped cell could be reproducibly generated [22]. Cells cultured on these crossbow shapes formed a lamellipodium at the broad, convex end whereas SFs formed along the sides and ventral portion of the cell. Thus, geometry influences organization of the actin cytoskeleton in a manner that influences cell polarization. Fibrous structures also change SF behaviour. Chao and colleagues [23] have cultured fibroblasts on electrospun poly-L-lactic acid fibres that model the waviness (crimp) of collagen-I fibres *in vivo*. Using this system, they showed that the degree of crimp affects the actin cytoskeleton: straight fibres result in cells having more actin SFs while increasing waviness of the fibres leads to shorter and thicker actin bundles [23].

# 5. Cell and stress fibre alignment on cylinders and the role of actin bending penalties

Studies dating back to the 1940s showed that Schwann cells cultured on glass fibres with 13 µm radii aligned along the cylinder axis [24]. A seminal paper by Dunn & Heath in 1976 [11] then showed that the degree of axial alignment is inversely related to cylinder radius, indicating that alignment on cylinders is dependent on the degree of micron-scale curvature. From these and related experiments, Dunn & Heath proposed a new hypothesis that actin microfilaments (newly discovered at that time) serve as structural elements and would be unable to properly operate when bent. In essence, they proposed that cells would preferentially align actin filaments, and ultimately the cell bodies themselves, to minimize the energy penalty from bending [11].

Work from other groups has reinforced the importance of actin bending penalties but also provided additional insight into SF bending in response to curvature. Svitkina *et al.* [25] showed that while most of the SFs in normal fibroblasts are long and align along the axial direction, fibroblasts also have a small population of short SFs that wrap in the circumferential direction, a condition in which SFs are the most bent. Additionally, some epithelial and transformed cells lack the curvature response of normal fibroblasts and have SFs aligned in the circumferential direction [25–28]. Thus, while the notion of actin bending penalties remains an important concept in understanding curvature cues, the mechanism(s) driving SF

alignment cannot be explained solely by an energy penalty for actin bending.

Consistent with the studies above, we found that isolated human VSMCs and mouse embryo fibroblasts (MEFs) align in the axial direction when cultured on cylinders with radii equal to the cell length scale and that this preferential alignment was lost as the radius of curvature increased [29]. We also observed the two SF subpopulations described by Svitkina *et al.* (see above) in these cells, but a Z-stack analysis showed that these two subpopulations could be distinguished by their position relative to the nucleus [29]. One population reached over the top of the nucleus, and we have called these 'apical' SFs. These apical SFs align strongly in the axial direction on cylinders with small radii of curvature and align less strongly as the radius of curvature is increased. The second, circumferentially aligned subpopulation was found below the nucleus, and we have called these 'basal' SFs. Basal SFs align circumferentially, nearly orthogonal to apical SFs and are significantly shorter than apical SFs. Both apical and basal SFs are likely subgroups within the ventral SF class, as defined by the presence of focal adhesions at both ends.

Interestingly, Rho activation in cells on cylinders led to the disassembly of apical SFs whereas the basal SFs became thick and robustly aligned circumferentially [29]. In fact, by activating Rho in confluent monolayers of VSMCs, we recapitulated the circumferential orientation of f-actin that is observed within blood vessels *in vivo*. Curiously, inhibition of Rho kinase, a kinase activated by Rho-GTP, also eliminated the apical population of SFs, and a small number of circumferentially oriented, thin, basal SFs were the only ones that remained in Rho kinase-inhibited cells [29]. One possible explanation for this paradoxical finding is that basal SFs may be more stable and resistant to perturbations of the Rho/ROCK pathway than apical SFs.

As a complement to the studies above, in which cells were cultured on the convex outer surface of cylinders, others have cultured cells on concave, semi-cylindrical grooves of negative mean curvature (equation (2.2)). Yip *et al.* reported that the negative mean curved wells drive elongation of both fibroblasts and MDCK (epithelial) cells [30]. The degree of elongation was inversely correlated to the radius of curvature in the fibroblasts but was found to be bi-phasic in the MDCK cells. An energy minimization model indicated that MDCK cell elongation is dependent on the balance between curvature-dependent intercellular adhesion (which inhibits elongation) and intracellular cortical actin bending (which promotes elongation). Broaders *et al.* [31] reported that negatively mean-curved channels (troughs, 100 µm wide), but not positively mean-curved channels (ridges) or planar surfaces, promote detachment of MDCK cell sheets. Both sets of experiments linked the effects of negative mean curvature to the Rho pathway. Using actin SFs and nuclear polarization as surrogates, Liu *et al.* showed that axial alignment is attenuated at cylinder ends [32], perhaps because the edge effect dominates over the curvature effect.

# 6. Cell and stress fibre alignment on more complex micron-scale curved substrata

As discussed above, the cylinder is a common biological geometry with zero Gaussian curvature. However, surfaces with more complex curvature fields and non-zero Gaussian

royalsocietypublishing.org/journal/rsob   *Open Biol.* **9**: 190155

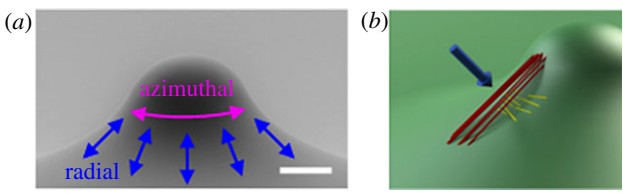

**Figure 3.** (*a*) Diagram of the sphere with skirt (SWS) showing the directions of radial and azimuthal migration. Scale bar, 50 μm. (*b*) Diagram of apical SFs forming chords (red) on the saddle region of the SWS versus the underlying basal SFs (yellow) which bend and align azimuthally. The blue arrow points in the direction normal to the surface at the centre of the cell. Images are reproduced with permission from [34]. Copyright © Elsevier.

curvatures exist throughout nature. Two recent studies probed the importance of non-zero Gaussian curvature substrata. Pieuchot *et al.* [33] fabricated a curved substratum in which two sinusoidal functions are mutually perpendicular to yield a surface of micron-scale 'hills' and bowl-like 'valleys' of defined dimensions. Cells plated on this surface see convex hills and concave valleys of positive Gaussian curvature smoothly connected via saddle-like negative Gaussian curvature domains. Recently, we fabricated a micron-scale substratum, called a sphere with skirt (SWS) that seamlessly connects a convex hill or 'spherical cap' of positive Gaussian curvature to a saddle-like skirt of negative Gaussian curvature [34] (figure 3*a*). The radius of curvature of the SWS surface (roughly 80–500 μm) is on the order of a cell length scale. These two platforms are complementary in that both surfaces contain micron-scale regions of positive and negative Gaussian curvature but differ in their concavity/convexity. The Pieuchot platform has both convex hills and concave bowl-shaped domains, whereas the SWS has convex spherical capped regions. On the SWS, isolated MEFs and VSMCs stably adhere and migrate on the surfaces of saddle-like curvature but are never observed on the spherical cap. Similarly, fibroblasts and mesenchymal stem cells avoid the hills and likely traverse the saddle-like domains in the Pieuchot platform to settle in the bowl-like valleys, a response they have termed 'curvotaxis'.

The selective adhesion of cells in bowls and on saddle-like surfaces might be related to the SF bending argument described above. Our live cell imaging of fluorescently-tagged actin in MEFs [34] showed that cells extend lamellipodia onto the spherical caps of the SWS, but the apical SFs terminate near the line of inflection (i.e. the line where the spherical cap transitions to the skirt). We posit that the cells are using these lamellipodia to probe their curvature landscape but find that they are unable to form sufficiently stable adhesions to support the SFs that would allow radial migration onto the cap. Thus, on a convex positive Gaussian curvature surface (i.e. the SWS spherical cap), there is no configuration in which the long, apical SFs could avoid bending. Consistent with this hypothesis, we found that MEFs were able to attach and spread on the spherical cap of a larger SWS surface with similar overall geometry but smaller principal curvatures. The apical SFs in these cells were significantly shorter than those in cells on the skirts of this surface, a finding that also supports the resistance to bending argument because the total energetic penalty for SF bending scales with the length of the SF.

A confocal scan through the cells revealed that apical and basal SFs are also present in cells cultured on the negative Gaussian curvature skirt portion of the SWS; apical SFs

align in the radial direction while basal SFs aligned largely in the azimuthal direction (figure 3*b*). Although the long, apical SFs on the SWS align along the direction of maximum curvature, they also remained unbent. Instead of following the surface contour, these SFs formed straight chords over the saddle-like portion of the surface (figure 3*b*). Thus, much like on cylindrical surfaces, apical SFs in cells on a saddle-like surface of negative Gaussian curvature align in a manner that minimizes their bending. This same argument may explain the ability of cells to settle in bowl-like valleys [33]. Basal SFs, on the other hand, preferentially aligned in the direction in which they are *most* bent on the SWS surface, again similar to the alignment behaviour we and others have seen in cells on cylinders [25,29].

Biton & Safran have developed a theoretical model [35] that may explain the unbent versus bent alignment patterns of apical and basal SFs, respectively, based on balance between SF bending energy and contractility. In this model, SFs tend to align in the axial direction when bending energy dominates. However, if contractility dominates over bending energy penalties, then SFs tend to align along the circumferential direction. Additionally, this argument has the potential to explain the enhanced appearance and circumferential alignment of basal SFs in Rho-activated cells on cylinders, as Rho activation increases myosin II-mediated contractility [36]. This model, however, does not fully account for the dynamic nature of SFs, which are continually polymerizing and depolymerizing as the cell migrates and proliferates. While bending energy penalties may adequately describe an inert material, such as a metal or glass rod, they do not fully encapsulate the behaviour of a complex network of polymers within a living cell.

Recently, Winkler *et al.* developed a physics-based reductionist model that simulated lamellipodial-based cell motility on curved surfaces in response to anisotropically aligned f-actin tangent and normal to the surfaces [37]. Cell alignment, migration velocity and direction were influenced by the principal curvatures and were related to the geometric features of the cell, the generation of actin filaments adjacent to the surfaces, and the associated forces exerted by these filaments. Several experimental findings were replicated with this computational model, including alignment and migration along a cylindrical axis when cells were seeded on the exterior of a cylinder. Additionally, the model can account for key features of migration on SWSs such as avoidance of spherical caps and repolarization of migration directions on surfaces with negative Gaussian curvature. The alignment and migration of elongated cells in this model did not include potential contributions of the cell nucleus nor did the model include actin SFs *per se*. Thus, the role of SFs, including possibly distinct effects arising from differential bending of apical versus basal SFs, remains to be addressed.

Others have also investigated the mechanics of cells and curvature [30,38–42]. These investigations provide insight into the manner in which cells respond to topography in the form of curvature, channels and grooves, as well as identify the cytoskeletal elements that play key roles in cellular responses to curvature. For instance, in fibroblasts, microtubules align with a grooved substrate first, followed by focal adhesions, then by SFs [40]. However, actin and microtubule disruption experiments performed in neurons by Hoffman-Kim *et al.* found that cell orientation was reduced

royalsocietypublishing.org/journal/rsob    Open Biol. 9: 190155

when actin was disrupted but not when microtubules were disrupted [38]. The difference in cell types notwithstanding, these results suggest that actin may be the more critical structure regarding cellular orientation on topographically varied substrata. Consistent with this idea, Pieuchot et al. reported that inhibition of actin polymerization, myosin-mediated contractility, Arp2, Cdc42 and Rho prevented curvotaxis whereas inhibition of microtubule polymerization with nocodozole had little effect [33]. Similarly, the perinuclear distribution pattern of vimentin intermediate filaments and the distribution of microtubules are relatively resistant to changes in micron-scale curvatures [43].

# 7. Micron-scale curvature drives differential gene expression

Micron-scale curvature can act as a microenvironmental cue governing gene expression. Chao et al. found that micron-scale waves generated from electrospun poly-L-lactic acid fibres direct fibroblasts to express high levels of type I collagen and tenascin C as compared to the same cells cultured on straight fibres; the effect of waviness on collagen-I mRNA levels could be reversed by stretching the fibres to 4% strain [23]. Soscia et al. cultured a parotid gland acinar cell line on micron-scale craters that modelled the micron-scale curvature of the basement membrane surrounding salivary gland acinar cells. The results showed that culturing acinar cells on craters of 30-μm radii led to increased expression of occludin and aquaporin-5 [44]. RNASeq analysis of mesenchymal stem cells on the sinusoidal curvature platform developed by Pieuchot et al. [33] identified several hundred genes that were differentially expressed by 5-day exposure to curvature; many of these genes were downregulated by the curvature cue. The downregulated genes include those that encode proteins expressed in differentiated tissues, transcription factors involved in differentiation processes, and genes involved in the response to stress, cytoskeleton remodelling, and cell proliferation. However, transcriptional effects selectively arising from negative and positive Gaussian regions of the saddle-like sides and bottom valleys, respectively, of the sinusoids were not individually explored [33].

How might substratum curvature regulate gene expression? Although this is probably a complex issue, one possible effect of concavity may be related to changes in nuclear compression. For example, when cells are cultured on planar surfaces, their apical SFs can make deep grooves in the surface of the nucleus; when these apical SFs are disrupted, compression is relieved and the nucleus can become up to approximately 60% thicker [45,46]. These types of changes in nuclear shape and size affect chromatin organization and gene expression [47–49]. Curvature-sensitive nuclear deformation can reduce mechanical restriction in nuclear pores and increase active nuclear import of the mechanosensitive transcriptional co-regulator, YAP [50]. Finally, indentation of the nucleus controls retention of DNA repair factors [51]. The formation of chords in cells cultured on surfaces with negative Gaussian curvature may stabilize cell adhesion while lessening the degree to which SFs indent the nucleus. And as discussed above, the same principle may apply to cells in bowl-like valleys. Changes in the transcriptome and genome integrity due to substratum curvature effects on SF-mediated nuclear compression provides a very new way of thinking about the spatial regulation of gene expression within cells and tissues.

# 8. Micron-scale curvature controls cell motility and nuclear positioning

Recent work has begun to establish how micron-scale curvature affects cell motility and underlying changes in SFs and nuclear positioning. Xu et al. [52] micropatterned curved strips that model airway walls of varying micron-scale curvatures and found a biphasic response in migration velocity, which first decreased and then increased with curvature. Larger curvature fields on the micron scale also reduce the rate at which epithelial sheets undergo collective migration inside microtubes [53] and increase the rate at which mesenchymal stem cells (MSCs) migrate on the outside of cylinders [54]. Moreover, when the radius of curvature roughly equals the cell length scale, MSCs preferentially migrate along a long cylinder axis (where apical SFs are less bent; see above) even if the adhesive ECM protein (collagen) is imprinted orthogonally. Thus, in this system, a micron-scale curvature cue dominates over a nanoscale ECM cue [54].

In our SWS surface, real-time imaging showed that MEFs migrate radially (up the platform towards the cap) through the region of negative Gaussian curvature until they reach the inflection point (figure 3b). At this point, they continue to explore their curvature microenvironment by extending and retracting protrusions but change their direction of migration to azimuthal, around the circumference of the feature [34]. Similarly, cells also extend and retract protrusions when plated on the sinusoidal platform developed by Pieuchot et al., but in this geometry, the cells prefer to keep their nuclei in the concave valleys and move them in a saltatory manner, presumably to minimize contact with convex surfaces during cell migration [33]. The nuclei in these valleys are more spherical than those on flat surfaces. Both azimuthal motility on the SWS inflection point and nuclear residence in the valleys of sinusoids are likely to minimize SF nuclear compression as discussed above.

In cylindrical grooves, formation of long, thick SFs is impeded by high curvature, and this effect may promote cell migration in the axial direction [54]. Interestingly in collectively migrating cells, leader and follower cells similarly aligned their SFs in the axial direction when exposed to high curvatures (diameters less than 75 μm), but the curvature cue was better retained in leader cells as the degree of curvature increased [53]. We found that the ratio of apical to basal SFs remains largely unchanged as MEFs migrate from near-planar to negative Gaussian regions of the SWS even though SFs alignment and direction of cell migration change as discussed above. This finding is very different from the behaviour of cells on planar and near-planar surfaces where apical SFs align in and predict the direction of cell migration [55].

Relationships between micron-scale curvature and nuclear positioning are just beginning to be explored. Chao and colleagues showed that changes in the degree of micron-scale crimp regulate nuclear orientation with strain [56]. Positioning of the nucleus, typically behind the centroid at least in cells on flat surfaces, is thought to be a prerequisite for migration and is strongly regulated by the subset of actin filaments that has been termed dorsal actin cables or perinuclear actin caps [18,46,57–59]. This actin subset connects to the LINC complex,

a multi-component complex containing outer nuclear membrane nesprins and inner nuclear membrane SUN proteins that connect the actin cytoskeleton to the nuclear lamina [59,60]. LINC complex disruption interferes with nuclear positioning and migration of cells on flat surfaces [57,58,61], as well as curvotaxis of cells on sinusoidal surfaces [33]. We posit that the uncoupling between the direction of apical SFs and migration on the SWS may also reflect a curvature-sensitive coupling and uncoupling between actin filaments and the nucleus via the LINC complex.

## 9. Can the principles of soft matter physics help to inform our understanding of curvature sensing by cells?

There are provocative correlations between the alignment of cells and inanimate soft matter in response to micron-scale curvature. For example, elongated colloids align radially and migrate when trapped at fluid–fluid interfaces shaped like a negative Gaussian curvature skirt [62]. Additionally, elongated block copolymer domains align radially on the skirt of a SWS-like surface [63]. In these soft matter systems, curvature alignment emerges as a result of free energy minimization. Elongated colloids align radially on negative Gaussian curvature interfaces to minimize the interfacial area and thus the interfacial energy. The cylindrical block copolymer domains are constrained to be equally spaced and adsorbed to the surface, but they also align radially on a negative Gaussian curvature surface to minimize the free energy of the system. Surface tension in the case of colloidal particles and bending energies in the case of cylindrical block copolymer domains impose energetic penalties for deviating from preferred

patterns. Reacting or time-evolving systems evolve along pathways to minimize their free energy, so related thermodynamic concepts apply even far from equilibrium.

Elongated cells and their actin SFs are reminiscent of these cylindrical block copolymer domains in that they are anisotropic in shape and resistant to bending. Even if cells and their apical SFs seek to minimize their free energy like the polymer system, the means by which they do this are probably very different. In the passive polymer system, thermal fluctuations enable the molecules to explore their energy landscapes and ultimately find minimum energy configurations. Instead of being driven by thermal fluctuations, cells actively explore free energy landscapes by consuming energy, typically by hydrolysing ATP and/or GTP. Is the consumption of these energy currencies driven by a complex decision-making process, or does it rather allow cells to move dynamically along paths that reduce their free energy (or equivalently maximize entropy production) as in inanimate, soft matter systems? The similarities between the alignment patterns observed in passive systems and cells suggest that at least a component of cell and SF alignment may be defined by energy minimization. Even as molecular mechanisms of curvature sensing become better elucidated, the degree to which principles of soft matter physics can guide cellular responses to the complex curvatures of developing and preexisting biological tissues will remain a fascinating matter for further study.

Data accessibility. This article has no additional data.

Competing interests. We declare we have no competing interests.

Funding. This work was supported by NIH grant no. HL137232 and the Center for Engineering MechanoBiology, a National Science Foundation Science and Technology Center, under grant agreement CMMI 1548571.

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
