## [Reviewer comments · Open Biology]

Review History

RSOB-19-0155.R0 (Original submission)

Review form: Reviewer 1

Recommendation

Accept with minor revision (please list in comments)

Do you have any ethical concerns with this paper?

No

Comments to the Author

In this manuscript, the authors summarized studies that intended to understand how cells respond to micron-scale surface curvatures in their microenvironment. The role of surface curvatures as essential physical cues to guide cell behaviors in biological systems has attracted lots of attention recently. This review will not only be an overview of this topic, but also provide insights into studies showing that cells can reorganize their cell bodies, actin fibers, nuclei and gene expression to adapt various micron-scale geometries. Overall, I believe this manuscript will be of interest to a wide audience and recommend the publication if the following issues are properly addressed by the authors.

Some comments

1. In Line 45, the authors pointed out that membrane proteins using BAR domains to interact with and create membrane curvature on the nano-scale has been reviewed. However, it has been well known that nano-scale curvatures interact with proteins beyond BAR domain proteins. Moreover, reference 1 and 2 both just used one section to talk about BAR domain proteins or nano-scale curvature. I suggest the authors revise this sentence and add references that mainly focus on nano-scale curvature or BAR domain proteins, e.g., Rao, Y. & Haucke, V. *Cell. Mol. Life Sci.* 68, 3983–3993 (2011) and Lou, H.-Y. et al. *Accounts of Chemical Research* 51, 1046–1053 (2018).
2. In Line 76, the title of this section is “Cell types impacted by micron-scale curvature”. However, the contents in this section can only imply that cell types and geometries are related. In other words, the audience may find that the studies presented here are insufficient to support such claim. I suggest rephrasing the title or adding new discussions into this section.
3. In the sections about actin reorganization and cell migration, I suggest the authors discuss one recent work describing how cells sense large-scale curvature and form anterograde actin flow at convex edges to switch migration mode (Chen, T. et al. *Nat. Phys.* 15, 393–402 (2019)). This very recent study should be noticed, as it is highly related to the topic of this manuscript.

Minor issues

1. For audiences who are not familiar with experimental details, the authors should avoid using technical terms without proper explanations. For example, in Line 198 and 210, what “LifeAct” and “phalloidin” stand for should be clarified.
2. In Line 263, the authors wrote “Using the fiber crimp system described above”. Most audiences would find this extremely confusing, as it is Line 115 where the fiber crimp system was mentioned last time. The authors should refer where exactly this system is described above or briefly describe this system again to help readers understand.
3. Since stress fibers are intensively discussed in this manuscript, I suggest the authors add one figure or more sentences to define different subtypes of stress fibers. Their general functions in cells should also be described.
4. The abbreviations (e.g., f-actin and MDCK) should always be defined where they first appear in the manuscript.
5. In Line 284, it is not accurate to state that curvature-sensitive nuclear deformation can also open the nuclear pore complex. In reference 50, their results only suggest that forces drive YAP nuclear translocation by reducing mechanical restriction in nuclear pores and increasing active nuclear import of YAP.

Decision letter (RSOB-19-0155.R0)

09-Sep-2019

Dear Dr Assoian

We are pleased to inform you that your manuscript RSOB-19-0155 entitled "Cellular sensing of micron-scale curvature: a frontier in understanding the microenvironment" has been accepted by the Editor for publication in *Open Biology*. The reviewer(s) have recommended publication, but also suggest some minor revisions to your manuscript. Therefore, we invite you to respond to the reviewer(s)' comments and revise your manuscript.

Please submit the revised version of your manuscript within 14 days. If you do not think you will be able to meet this date please let us know immediately and we can extend this deadline for you.

- 1) A text file of the manuscript (doc, txt, rtf or tex), including the references, tables (including captions) and figure captions. Please remove any tracked changes from the text before submission. PDF files are not an accepted format for the "Main Document".
- 2) A separate electronic file of each figure (tiff, EPS or print-quality PDF preferred). The format should be produced directly from original creation package, or original software format. Please note that PowerPoint files are not accepted.
- 3) Electronic supplementary material: this should be contained in a separate file from the main text and meet our ESM criteria (see <http://royalsocietypublishing.org/instructions-authors#question5>). All supplementary materials accompanying an accepted article will be treated as in their final form. They will be published alongside the paper on the journal website and posted on the online figshare repository. Files on figshare will be made available approximately one week before the accompanying article so that the supplementary material can be attributed a unique DOI.

Online supplementary material will also carry the title and description provided during submission, so please ensure these are accurate and informative. Note that the Royal Society will not edit or typeset supplementary material and it will be hosted as provided. Please ensure that the supplementary material includes the paper details (authors, title, journal name, article DOI). Your article DOI will be 10.1098/rsob.2016[last 4 digits of e.g. 10.1098/rsob.20160049].

- 4) A media summary: a short non-technical summary (up to 100 words) of the key findings/importance of your manuscript. Please try to write in simple English, avoid jargon, explain the importance of the topic, outline the main implications and describe why this topic is newsworthy.

Images

Data-Sharing

It is a condition of publication that data supporting your paper are made available. Data should be made available either in the electronic supplementary material or through an appropriate

repository. Details of how to access data should be included in your paper. Please see <http://royalsocietypublishing.org/site/authors/policy.xhtml#question6> for more details.

Data accessibility section

To ensure archived data are available to readers, authors should include a ‘data accessibility’ section immediately after the acknowledgements section. This should list the database and accession number for all data from the article that has been made publicly available, for instance:

Sincerely,
The Open Biology Team
<mailto:openbiology@royalsociety.org>

Reviewer(s)' Comments to Author:

Referee: 1

Comments to the Author(s)

In this manuscript, the authors summarized studies that intended to understand how cells respond to micron-scale surface curvatures in their microenvironment. The role of surface curvatures as essential physical cues to guide cell behaviors in biological systems has attracted lots of attention recently. This review will not only be an overview of this topic, but also provide insights into studies showing that cells can reorganize their cell bodies, actin fibers, nuclei and gene expression to adapt various micron-scale geometries. Overall, I believe this manuscript will be of interest to a wide audience and recommend the publication if the following issues are properly addressed by the authors.

Some comments

1. In Line 45, the authors pointed out that membrane proteins using BAR domains to interact with and create membrane curvature on the nano-scale has been reviewed. However, it has been well known that nano-scale curvatures interact with proteins beyond BAR domain proteins. Moreover, reference 1 and 2 both just used one section to talk about BAR domain proteins or nano-scale curvature. I suggest the authors revise this sentence and add references that mainly focus on nano-scale curvature or BAR domain proteins, e.g., Rao, Y. & Haucke, V. *Cell. Mol. Life Sci.* 68, 3983–3993 (2011) and Lou, H.-Y. et al. *Accounts of Chemical Research* 51, 1046–1053 (2018).
2. In Line 76, the title of this section is “Cell types impacted by micron-scale curvature”. However, the contents in this section can only imply that cell types and geometries are related. In other words, the audience may find that the studies presented here are insufficient to support such claim. I suggest rephrasing the title or adding new discussions into this section.
3. In the sections about actin reorganization and cell migration, I suggest the authors discuss one recent work describing how cells sense large-scale curvature and form anterograde actin flow at convex edges to switch migration mode (Chen, T. et al. *Nat. Phys.* 15, 393–402 (2019)). This very recent study should be noticed, as it is highly related to the topic of this manuscript.

Minor issues

1. For audiences who are not familiar with experimental details, the authors should avoid using technical terms without proper explanations. For example, in Line 198 and 210, what “LifeAct” and “phalloidin” stand for should be clarified.
2. In Line 263, the authors wrote “Using the fiber crimp system described above”. Most audiences would find this extremely confusing, as it is Line 115 where the fiber crimp system was mentioned last time. The authors should refer where exactly this system is described above or briefly describe this system again to help readers understand.
3. Since stress fibers are intensively discussed in this manuscript, I suggest the authors add one figure or more sentences to define different subtypes of stress fibers. Their general functions in cells should also be described.
4. The abbreviations (e.g., f-actin and MDCK) should always be defined where they first appear in the manuscript.
5. In Line 284, it is not accurate to state that curvature-sensitive nuclear deformation can also open the nuclear pore complex. In reference 50, their results only suggest that forces drive YAP nuclear translocation by reducing mechanical restriction in nuclear pores and increasing active nuclear import of YAP.

Author's Response to Decision Letter for (RSOB-19-0155.R0)

See Appendix A.

Decision letter (RSOB-19-0155.R1)

24-Sep-2019

Dear Dr Assoian

We are pleased to inform you that your manuscript entitled "Cellular sensing of micron-scale curvature: a frontier in understanding the microenvironment" has been accepted by the Editor for publication in Open Biology.

Sincerely,

The Open Biology Team
mailto: openbiology@royalsociety.org

Appendix A

We thank the editor and reviewer for the careful review of our manuscript. Our replies to the reviewer comments follow.

1. We have replaced the two questioned references (references 1 and 2) with the two more robust references suggested by the reviewer.
2. We have changed the title of the subsection as requested by the reviewer (line 75 of the revised manuscript).
3. We have incorporated the suggested study by Chen et al. (lines 115-118 and reference 21 of the revised manuscript).

Minor issues

1. We have eliminated the technical terms from the revised manuscript as requested by the reviewer.
2. As requested, we have clarified the language describing the Chao et al. system (lines 270-273 of the revised manuscript).
3. We have added several sentences describing stress fibers as requested by the reviewer (lines 102-107 of the revised manuscript).
4. We have tried to define all abbreviations at first use as requested.
5. We have rephrased the statement in accordance with the reviewer's suggestion (lines 292-293 of the revised manuscript).

Sincerely,
Richard Assoian